# "We are left with nothing to work with"; challenges of nurses working in the emergency unit at a secondary referral hospital: A descriptive qualitative study

Agani Afaya[1,2]*, Victoria Bam[3], Thomas Bavo Azongo[4], Richard Adongo Afaya[5], Vida Nyagre Yakong[5], George Kwame Kpodo[2], Robert Alhassan Kaba[6], Denis Albanus Nangsire Zinle[2], Daniel Kofi Tayuu[2], Stella Asantewaa[2], Peter Adatara[2]

1 College of Nursing, Yonsei University, Seoul, Republic of Korea, 2 Department of Nursing, School of Nursing and Midwifery, University of Health and Allied Sciences, Ho, Ghana, 3 Department of Nursing, Kwame Nkrumah University of Science and Technology, Kumasi, Ghana, 4 Department of Public Health, School of Allied Health Sciences, University for Development Studies, Tamale, Ghana, 5 School of Nursing and Midwifery, University for Development Studies, Tamale, Ghana, 6 Centre for Health Policy and Implementation Research, Institute of Health Research University of Health and Allied Sciences, Ho, Ghana

* aagani@uhas.edu.gh

## Abstract

### Introduction

In recent times, there has been an increasing burden in traumatic, medical, and surgical emergency conditions, placing more emphasis on the need for quality emergency care. This study aimed to explore the challenges experienced by nurses working in the emergency unit of a secondary referral hospital.

### Methods

The study used an exploratory qualitative research design with a constructivist approach and a grounded theory method. Data were collected through in-depth interviews lasting between 30 to 45 minutes using a semi-structured interview guide. Inductive content analysis was used to analyse data.

### Results

Eleven (11) participants were interviewed. The majority were female (9), aged between 31–40 years. From the inductive content analysis, four themes emerged. These were; 1) overcrowding in the emergency unit, 2) understaffing at the emergency unit, 3) lack of emergency equipment, 4) inadequate managerial support.

### Conclusion

The study identified several major challenges confronting nurses working in the emergency unit which are linked with managerial processes and inadequate managerial support. These challenges need to be addressed to promote quality emergency nursing care. To foster a

**Funding:** The author(s) received no specific funding for this work.

**Competing interests:** The authors have declared that no competing interests exist.

positive working environment, hospital management should validate and address the afore-mentioned concerns of the Emergency Department nurses.

## Introduction

Emergency care is an important component of every health system. In recent times, there has been an increasing burden in traumatic, medical, and surgical emergency conditions across the globe, placing more emphasis on the need for quality emergency care [1]. The World Health Organization (WHO) identified sub-Saharan Africa (SSA) with 1.3% of the world's healthcare workers and 25% of the global disease burden [2]. Unfortunately, this burden of emergency care is high in low-resource countries compared with the developed world, with developing countries having deficiencies in organisational planning, shortage of trained healthcare personnel, and inadequate resources to treat emergency conditions [1,3–5]. It is estimated that more than 90% of injury-related deaths occur in low-resource countries [6], where the burden of injury presents a major public health threat, highlighting the need for skilled healthcare personnel in the Emergency Departments (ED) to help avert preventable deaths.

Emergency care is important for the immediate management and stabilisation of critically ill patients. Although long overlooked as an essential component of healthcare in SSA, emergency health services are now growing across many countries. Emergency nursing is a specialty within professional nursing whereby the nurse provides care to patients requiring immediate medical attention to avoid long-term disability or death [2]. The emergency nurse plays a critical role in the identification and management of patients with life-threatening conditions, prioritises emergency care through triaging, performs resuscitation with appropriate management within a supportive health care setting [2,7].

Due to the shortage of nursing staff, limited specialty training, overwhelming patient numbers, and stressful working environments, the emergency nursing role especially in Africa is particularly challenging [8,9].

In the provision of quality healthcare across Africa, nurses play a central role. Therefore, it is imperative to increase the skilled nursing workforce and provide specialty emergency nursing training across SSA. In response to this, the international community has made efforts in low and middle-income countries to develop capacity and quality improvement programs for emergency care [1,10] and Ghana is not an exception. Most recent studies have called for increased attention to trauma and emergency care, but the health sector in Ghana currently has limited training institutions for these areas, resulting in delays in emergency care, and associated with poor clinical outcomes [7]. In 2012, Ghana started a degree programme in emergency nursing to help bridge the gap of lack of specialist emergency care nurses in the country. Since the introduction of the degree in emergency nursing programme in Ghana, there has been an increase in the number of emergency nurses in the country, however, most emergency units in hospitals across the country continue to have inadequate or no specialist emergency care nurses. A study conducted on the experiences of registered nurses in the emergency centres in the Volta region of Ghana identified some challenges confronting nurses working in the ED [11]. They found that resource challenges such as lack of space and inadequate emergency specialists endangers quality emergency care [11]. It is therefore prudent to explore the challenges of ED nurses to provide recommendations for improvement in the emergency care system in Ghana. This study aimed to

explore the challenges experienced by nurses working in the emergency unit of a secondary referral hospital.

## Methods

### Study design

The study used an exploratory qualitative research design with a constructivist grounded theory approach [12]. An interpretative approach [13] was employed to gain an in-depth understanding of the challenges that nurses face during emergency care delivery. In undertaking a constructivist survey the researchers adopted a position of mutuality where there was a relationship with participants that enabled a mutual construction of meaning during interviews and a meaningful reconstruction of their stories into a grounded theory model [12,14].

### Research team and reflexivity

The researchers are professional nurses with much interest in emergency nursing care. The interviews were conducted by either AA, GKK, DANZ, DKT, or SA who all had experience in conducting qualitative studies. AA and RAA had a previous interviewing experience during their Masters' studies and several other qualitative research projects. GKK, DANZ, DKT, and SA are clinical nurse researchers. VB has a PhD and conducts research in emergency nursing education. TBA, VNY, RKA, and PA holds PhDs and have special interest in emergency nursing research.

The current study adopted two forms of reflexivity in qualitative research known as personal reflexivity and epistemological reflexivity. With regards to personal reflexivity, all the research team members found themselves ruminating on how their experiences in nursing practice especially emergency nursing might shape the discussions. Authors also pondered on how the findings might have affected them as educators and professionals. All the participants were known to one of the interviewers (GKK) and some were known by the interviewers through workshops and professional networks. On the issue of epistemological reflexivity, the researchers were faced with a dilemma relating to the study methodology. The researchers proposed to combine both individual and focus group interviews but due to the busy nature of the ED, the researchers adopted individual interviews. The individual interviews provided the opportunity for participants to freely share detailed and sensitive information regarding their challenges at the ED.

### Study setting and population

This research was conducted at the emergency unit of a secondary referral hospital in the Ho municipality. It is one of the Ghana Health Services' (GHS) facilities situated in the Volta region of Ghana. The emergency unit is a small unit attached to the outpatient department (OPD) that provides twenty-four-hour emergency services to clients. It admits and detains patients for not more than twenty-four hours. The emergency unit had two beds. The unit also utilised a couch and five stretchers as beds.

The participants for this study were registered general nurses and clinical nurse assistants with active professional identification numbers (PINs) and auxiliary identification numbers (AINs) respectively who have been working in the emergency unit for a period not less than six (6) months. At the time of the study, the emergency unit had no emergency nurse specialist. There were two senior nursing officers, a nursing officer, a senior staff nurse, four staff nurses, and four clinical nurse assistants working in the unit. The nurses in the ED had a six-hour shift system for day duty while the night duty lasted for 12 hours.

## Sampling and data collection procedure

Participants were selected through purposive sampling. In qualitative research, purposeful sampling is a technique that is widely used for the identification and selection of information-rich cases for the most effective use of limited resources [15]. This involves identifying and selecting participants who are available and willing to participate and should be able to communicate their experiences and opinions in an expressive, articulate, and reflective manner [16–18]. A written and verbal consent was obtained from participants after the purpose of the study was explained to them. The participants were recruited after their daily duty for the interview. Those who had busy schedules after their daily duty were approached on a different day to undertake the interview. Data were collected through in-depth interviews lasting between 30 to 45 minutes using a semi-structured interview guide. The interview guide was developed based on review of related literature. The interview guide was pretested and all necessary corrections were made before commencing the interviews. Face-to-face individual interviews were conducted at the emergency unit of the study hospital in a secluded office. Due to the busy nature of the unit, the authors had to arrange a convenient time with participants who were willing to participate in the study. Questions were asked about emergency care delivery and the challenges experienced during care delivery in the unit. Prompts and probing questions were used for further information and explanation following participants' responses. During the interview sessions, the researchers provided an enabling environment for participants to express themselves freely without interferences, and leading questions were avoided to ensure that participants were not interrupted when sharing their experiences. Data saturation was reached at the eleventh participant when no new information was obtainable. All interviews were digitally recorded and transcribed verbatim by GKK, DANZ, DKT, and SA. Audio files and transcripts were stored anonymously in a secured protected digital storage system.

## Data analysis

The study employed qualitative content analysis to analyse data as described by Padgett [19]. The appropriate form of analysis was inductive content analysis because there were no pre-empted themes to guide the study. The researchers read and reread the transcripts several times to make meaning of participants' views/challenges. The first author (AA) and sixth author (GKK) generated an independent coding frame through reading and re-reading the interview transcripts focusing on keywords, sentences, and phrases to reduce data to codes. AA and GKK coded the same interviews to identify and discuss the differences in coding to check for intercoder reliability. The generated codes were constantly compared for conceptual similarity and differences by referring to the original interview transcripts. Similar codes were combined into themes. An iterative process was used during the revision of codes, through backward and forward data assessment and analysis resulting in the verification and modification of themes. Regular data sessions were organised in which all the authors met with AA and GKK to review the coding and mutually agreed on the codes and then reached a consensus on how to form the final themes.

## Trustworthiness

To ensure trustworthiness, the researchers applied the principles of credibility, dependability, confirmability, and transferability [20]. Member checks were done at the end of each interview to ensure participants' views were well presented ensuring credibility. The researchers provided a detailed description of the study setting, methodology (COREQ criteria were used) [21], and background of the study sample to allow for the transferability of the findings in a

similar context and setting. To ensure confirmability, an audit trail was kept for other researchers to validate the processes undertaken in the study. The researchers' ensured dependability through peer debriefing with all the research activities and processes described in detail to enable replication.

## Ethical consideration

Ethics approval to conduct this study was obtained from the University of Health and Allied Sciences Research Ethics Committee (**UHAS-REC A.9 [5] 18–19**) Ho, Ghana. Permission was sought from the hospital management and unit manager before the commencement of data collection. Potential participants were informed about the purpose of the study. Participants who were approached to participate in the study all gave verbal and written consent. They were informed about their right to withdraw from the study at any particular point in time. Participants were assured of no harm during and after the data collection process. Confidentiality, privacy, and anonymity were ensured by complying with the Data Protection Act [22] and not letting others know who has and has not participated in the study, and also ensuring participants are not identifiable via other information. To further ensure these, random codes were given to participants to maintain anonymity due to the smaller number of nurses in the unit. Continuous codes were easy to trace therefore the researchers adopted random codes. These clouded participants' from knowing who was either interviewed before or after with the code numbers.

## Findings

**Participants' characteristics.**   Eleven participants were interviewed including 8 registered general nurses and 3 clinical nurse assistants. The majority of the participants were women (9) and aged between 31–40 years. Most of the participants (7) had more than 3 years of working experience in the emergency unit (Table 1).

Four themes emerged from the data which were; 1) overcrowding in the emergency unit, 2) understaffing at the emergency unit, 3) lack of emergency equipment, 4) inadequate managerial support.

*Theme 1*: **Overcrowding in the emergency unit.**   Overcrowding in the emergency unit was a great concern for participants. Participants expressed that inadequate space in the unit hinders quality nursing care delivery. Participants described the emergency unit as a 'chamber'

**Table 1. Participants characteristics.**

| Characteristics | | Number |
|---|---|---|
| Total | | 11 |
| **Gender** | Male | 2 |
| | Female | 9 |
| **Age** | 21–30 | 2 |
| | 31–40 | 9 |
| **Academic Qualification** | Certificate | 3 |
| | Diploma | 5 |
| | Bachelors | 3 |
| **Nursing Category** | Clinical Nurse Assistant | 3 |
| | Registered General Nurse | 8 |
| **Working Experience** | 6months-3years | 3 |
| | 4years-7years | 6 |
| | >7years | 2 |

meaning it is too small to be used as an emergency ward. Others described the ward as very hot due to poor ventilation.

*"Currently the bed capacity in the emergency unit is only two but you can sometimes have as many as five to eight patients at the same time and some may end up sitting down or we use worn-out stretchers as improvised beds that are not comfortable for them to receive their treatments. The overcrowding caused by limited beds and space makes it very difficult for us to move between patients to deliver care. We are at risk because anything can happen to us as caregivers."* (Participant 129).

Almost all the participants expressed their frustration in having to nurse patients on wheelchairs and stretchers making the performance of some nursing procedures practically impossible. Participants reported not performing the full iterative circle of the nursing process due to an unconducive working environment. Manoeuvring through defective stretchers, wheelchairs, and benches in the delivery of quality emergency care is frustrating and poses hazards to both the nurse and the patient.

*". . .at times, we can have fifteen patients at once, and we have two beds in the emergency room and there are times that we have to nurse patients in chairs. . .and this makes work very difficult"* (Participant 127).

*". . .the stretchers, wheelchairs with the beds make the place so congested. . .the environment is very small and the way many patients are rushed in makes the place not conducive"* (Participant 125).

***Theme 2*: Understaffing at the emergency unit.** Understaffing was a major issue expressed by all participants. According to the participants, adequate staffing is necessary for effective emergency care and reduces work stress. Participants indicated that in some cases, nurses have to run shifts alone or with students amidst increasing workloads. They further emphasized that it was more worrying when clinical nurse assistants with limited scope of practice and required to work under the supervision of a registered nurse were allowed to run shifts alone. This practice has implications for the provision of quality emergency nursing care to patients. Participants reported being overwhelmed with the increasing number of patients admitted to the unit and this affects healthcare delivery. In some cases, one nurse attends to about 6 cases per shift.

*"In terms of staff strength, we are lacking a lot to the extent that sometimes only one staff works in the emergency unit per shift"* (Participant 212).

Another participant said;

*". . .there is no trained emergency nurse in the unit. Staffing challenges make work very tedious because, work that four (4) or five (5) people should do, it is left to one person. This makes nursing care very difficult and leads to exhaustion. This leads to attention deficit and inadequate care for the client. Our emergency is very busy, we work 24/7 and you could manage about twenty patients during the day shift and I am always very tired when I get home"* (Participant 127).

***Theme 3*: Lack of emergency equipment.** Participants stated that the lack of some logistics and consumables at the emergency unit affected quality emergency care. Participants

mentioned that due to the influx of patients to the emergency unit, the two-bed capacity ward is inadequate to contain the patients. The wheelchairs and stretchers are all in a deplorable state as most are worn out. Some vital resources needed for delivery of care were sometimes not available at the time of need. Nurses often use their bare hands in performing invasive procedures or wait until patients' relatives purchase some of these resources or run to other wards to borrow them. The following are narrations from some participants.

*"...We have two BP [blood pressure] apparatuses that we share with the general Out Patient Department [OPD], so when a case comes to the emergency unit, you have to run to the OPD for the BP apparatus and the general OPD is on hold or chasing you back for the same apparatus. And the patient looks at you, that, you don't know what you are doing"* (Participant 127).

*"Sometimes unavailability of emergency drugs brings a lot of problems in the emergency unit...we are left with nothing to work with..."* (Participant 212).

Most nurses working in the ED were left in a state of frustration as the most vital equipment needed to manage critical cases were either broken down or unavailable.

*"... sometimes patient report to the emergency and both the hospital and emergency stock is finished and we have to use our own money to buy common intravenous fluid [IVF] giving set. There are no glucometer strips to even check the patient's random blood sugar [RBS] when they come in collapsed. You either run to the wards for glucometer strips or send a relative to buy them outside. We sometimes don't have gloves or syringes to work with and have to use our bare hands for most of the invasive procedures. You have situations where there is no oxygen in the whole hospital and you are found wanting in front of a client who needs oxygen..."* (Participant 311).

*"... sometimes common disposable gloves are not available as well as safety boxes and it exposes us to a lot of occupational risks. You will run everywhere for oxygen just because there is no oxygen in the hospital and at the end of the day, you lose your client"* (Participant 129).

*Theme 4*: **Inadequate managerial support.** To deliver quality emergency care, overwhelmingly all participants pointed to the need to be supported and given the needed attention by management. The participants spoke at length about the lack of support by management and how this contributes negatively to their ability to provide quality care to clients. Some nurses singled out their unit head (in-charge) as being a supportive and motivating figure for them. They mentioned that the nursing administration does not give support whenever they have challenges in the unit. Most of the administrative issues were about the ineffectiveness of nursing administration-lack of effective listening and response to nurses' plight, lack of motivation, demotivating actions, and utterances. Some participants indicated that lack of motivation from the hospital's general administration and lack of proactive effort by the hospital to continuously supply emergency drugs and other vital equipment for emergency care was a major challenge to quality emergency care delivery. All participants awaited management to validate their work-related concerns and find appropriate solutions to improve quality emergency care in the unit. The narratives below were the most pervasive among the participants.

*"It's very difficult to get our superiors to understand what we are facing in the emergency unit. Immediately they come and the place is busy, you will hear them say, 'this place is heavy ooo, and they will disappear'. I wish that they will come, and work with us and understand what*

*we have been going through so that they won't be saying that we are always complaining. This challenge is always with the nursing administration because the medical superintendent is always with us and he understands what we go through. For our nursing administration, they always say we talk or complain too much, and it has got to a time where everybody says they will do what they can, and this is affecting the quality of care given to client and job satisfaction"* (Participant 127).

*". . .nursing administration doesn't understand the plight of the nurses working in the emergency unit. When we complain to them about consumables and logistics, they will shout at you saying that you complain too much. Meanwhile, the client is in front of you and you can't circumvent the problem. They like using this phrase- 'you're misusing the things'. Nursing administration doesn't give us listening ears and even if they do, they will say—'during my time it was like this'. We do what we can because our managers are not ready to support us. They tag you when you are trying to push for the best interest of the client"* (Participant 311).

*"Over the years nobody had recognized me for the works I've been doing- not even a handshake. If you do a good job, you need to be recognized and appreciated but that never happens but they're very quick to chastise you for a little mistake done. This doesn't motivate me to give my all"* (Participant 121).

Most participants after working in the emergency unit for many years believe that the hospital has no plans for nurses to go for further training in emergency care nursing, and to them, this would enhance their knowledge and skill in caring for patients requiring emergency medical intervention. They bemoaned that currently, the emergency unit has no trained emergency nurse specialist which is a major concern for all participants. Lack of continuing professional education was seen to be endangering the future of emergency nursing practice in the hospital.

*". . .I'm not aware of any policy of the hospital concerning the upgrading of nurses and nurses working at the emergency unit. Sometimes colleague nurses may do presentations on conditions and whoever is asked to do this presentation, research, and come and present. This is not effective; we need specialist training in emergency care nursing. . ."* (Participant 129).

## Discussion

The findings of this study highlight challenges confronting nurses working in the emergency unit. Amid these challenges, ED nurses continue to deliver emergency care whiles calling for their concerns to be heard and addressed to enhance quality emergency care delivery.

Overcrowding in the emergency unit was one of the major challenges expressed by nurses working in the ED, making healthcare delivery tedious. Nurses could not perform the full iterative circle of the nursing process due to the unconducive working environment and this affects the quality of emergency care delivery leading to poor patient outcomes. The current study finding is consistent with previous studies conducted across the globe identifying overcrowding in the ED as a major challenge to quality emergency care delivery [23–30]. Overcrowding in the ED may occur as a result of the shortage of nursing staff, the number of patients waiting to be seen, bed shortage, delays in treating or assessing patients already in the ED, or discharged patients yet to settle their bills and detained at ED [23,29,31,32]. Therefore, if the global crisis [33] of ED crowding is to be solved, interventions designed to address the problem must be tailored towards identified specific causes. We recommend that hospital management provide a spacious ED and also increase the bed capacity to contain the

increasing numbers of patients. The ED should further strengthen their triaging system by providing a triaging room at the ED as this will help identify acutely ill patients needing emergency care from non-critically ill patients who could be referred to the general ward. This will further prevent delays in referring and discharging patients from the ED to ease crowding.

The shortage of nursing staff in the ED remains an important issue of interest to hospital administrators, nurses, and physicians. The current study revealed that understaffing was a challenge experienced by ED nurses. According to the participants, adequate staffing is necessary for effective emergency care delivery and will improve key ED throughput metrics. To help bridge the gap most participants had to work overtime to assist their colleagues in the next shift. The findings of this study are not different from a Canadian study where participants expressed a shortage of nursing staff in the ED which led to ED nurses working overtime [34]. A similar challenge was found in China [35] where they faced a shortage of emergency nursing staff in the ED. The World Health Report in 2006 classified Ghana among 36 countries in SSA facing a health workforce crisis [36] which necessitated government interventions through the Ministry of Health to address a myriad health workforce challenges. Recently out of about 115 650 employed public sector health workers, 58% were nurses and midwives in Ghana [37,38]. From 2008 to 2018, there has been a remarkable increase in the nursing workforce of about 370% [37]. Nonetheless, empirically, there exists a persistent shortage of nurses and midwives in most hospitals and clinics across Ghana [37,38]. As in recent times over 40,000 trained nurses and midwives have been unemployed between 2016 to 2019 whiles health facilities across the country are in dire need of these professionals. Due to the extended credit facility agreement between the government of Ghana and the International Monetary Fund with its associated austerity measures, most of the graduated nurses from 2016 and 2018 could not be immediately employed [38].

Nurses in the ED recounted the challenges they encounter in delivering emergency care amid the unavailability of basic emergency equipment. In most emergency units in Ghana, patients reporting to the units are not triaged and most emergency centres are poorly equipped, under-resourced, and overcrowded [11]. Our study finding is consistent with a study conducted in Ghana by Atakro et al., [23] where nurses complained of a lack of material resources to work with. Another study by Hines, Fraze, and Stocks [39] also found that ED nurses were confronted with several challenges which included inadequate resources to work with due to decreased reimbursement by insurers. Participants in the current study expressed their dissatisfaction when they lack adequate resources to work with. As the ED is most often the first point of call for patients, resourcing it implies quality care, quick recovery and discharge, client and relative satisfaction, good hospital image, and reducing overcrowding.

To deliver quality emergency care, overwhelmingly all participants pointed to the need to be supported and given the needed attention by management. The participants articulated at length about management's lack of support and how this contributes negatively to their ability to provide quality care for their clients. This finding is consistent with a study conducted in Canada by Enns and Sawatzky [34]. The participants in the current study yearned for management to validate their work-related concerns and find appropriate solutions to improve quality emergency care in the unit. This finding is also congruent with the study of Enns and Sawatzky [34] in Canada.

Nurses in Ghana are trained in emergency care largely as general nurses with limited emergency care knowledge and skills. Participants in the current study have inadequate specialist training in emergency nursing. Participants cautioned that lack of professional continuing education in emergency nursing for nurses in the unit will endanger the future of emergency nursing practice in the hospital and the country as a whole. It is therefore imperative to develop career path or progression policies that reward increasing clinical competency, knowledge, education, and professional development for nurses.

## Strengths and limitations

The findings of this study are limited to the participants interviewed and to their challenges experienced in the emergency unit. Nonetheless, the rich exploration of the challenges affecting emergency care provides an opportunity for stakeholders to find solutions to address these concerns raised by ED nurses to improve quality emergency care.

## Conclusion

The study identified several major challenges confronting nurses working in the emergency unit such as overcrowding, understaffing, lack of resources, and limited managerial support. These challenges if not managed and resolved will endanger quality emergency nursing care delivery.

It is important that the government, and hospital managers pay attention and invest resources into EDs to reduce the challenges of inadequate resources. There is a need for hospital managers to develop innovative strategies and policies that will support the working environment of ED nurses to provide quality emergency care to patients. It is also imperative for management to validates the expressions of concerns from ED nurses and provide appropriate solutions to foster a positive working environment.

## Supporting information

**S1 File.**
(DOCX)

## Acknowledgments

The authors are grateful to the participants of this study.

## Author Contributions

**Conceptualization:** Agani Afaya, George Kwame Kpodo, Denis Albanus Nangsire Zinle, Daniel Kofi Tayuu, Stella Asantewaa.

**Data curation:** Agani Afaya, George Kwame Kpodo, Denis Albanus Nangsire Zinle, Daniel Kofi Tayuu, Stella Asantewaa.

**Formal analysis:** Agani Afaya, Richard Adongo Afaya, Vida Nyagre Yakong, George Kwame Kpodo, Robert Alhassan Kaba, Denis Albanus Nangsire Zinle, Peter Adatara.

**Methodology:** Agani Afaya, Victoria Bam, Thomas Bavo Azongo, Richard Adongo Afaya, Vida Nyagre Yakong, George Kwame Kpodo, Robert Alhassan Kaba, Denis Albanus Nangsire Zinle, Daniel Kofi Tayuu, Stella Asantewaa, Peter Adatara.

**Resources:** Agani Afaya, George Kwame Kpodo, Denis Albanus Nangsire Zinle, Daniel Kofi Tayuu, Stella Asantewaa.

**Supervision:** Agani Afaya, Victoria Bam, Thomas Bavo Azongo, Vida Nyagre Yakong, Robert Alhassan Kaba, Peter Adatara.

**Validation:** Agani Afaya, Victoria Bam, Thomas Bavo Azongo, Richard Adongo Afaya, Vida Nyagre Yakong, Robert Alhassan Kaba, Peter Adatara.

**Writing – original draft:** Agani Afaya, Richard Adongo Afaya, George Kwame Kpodo, Denis Albanus Nangsire Zinle.

**Writing – review & editing:** Agani Afaya, Victoria Bam, Thomas Bavo Azongo, Richard Adongo Afaya, Vida Nyagre Yakong, George Kwame Kpodo, Robert Alhassan Kaba, Denis Albanus Nangsire Zinle, Daniel Kofi Tayuu, Stella Asantewaa, Peter Adatara.

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
