## [Decision Letter · Decision Letter 0]

4 Dec 2020

PONE-D-20-22261

“We are left with nothing to work with”; challenges of nurses working in the emergency unit at a secondary referral hospital: a descriptive qualitative study

PLOS ONE

Dear Dr. Afaya,

Thank you for submitting your manuscript to PLOS ONE. After careful consideration, we feel that it has merit but does not fully meet PLOS ONE’s publication criteria as it currently stands. Therefore, we invite you to submit a revised version of the manuscript that addresses the points raised during the review process.

Please pay particular attention to the comments added to your pdf manuscript by Reviewer#1 which will strengthen your paper.

We look forward to receiving your revised manuscript.

Kind regards,

Katie MacLure, PhD, MSc (dist)., BSc (1st)

Academic Editor

PLOS ONE

Journal Requirements:

2. Please include a copy of the interview guide used in the study, in both the original language and English, as Supporting Information, or include a citation if it has been published previously.

Reviewers' comments:

Reviewer's Responses to Questions

**Comments to the Author**

1. Is the manuscript technically sound, and do the data support the conclusions?

Reviewer #1: Partly

Reviewer #2: Yes

2. Has the statistical analysis been performed appropriately and rigorously? 

Reviewer #1: I Don't Know

Reviewer #2: Yes

3. Have the authors made all data underlying the findings in their manuscript fully available?

Reviewer #1: No

Reviewer #2: Yes

4. Is the manuscript presented in an intelligible fashion and written in standard English?

Reviewer #1: No

Reviewer #2: Yes

5. Review Comments to the Author

Reviewer #1: Thank you for submitting this paper which I have read with interest and enjoyment. You and your colleagues have addressed an important area with implications for practice and patient safety. I think you’ve gathered very interesting data which I’m sure will add to an important debate.

I have made many comments which I hope will be helpful. Some contextualising information in the introduction section would be helpful for example how patients are triaged in the ED. I am concerned that the hospital may be identifiable and suggest that you anonymise it further – not just a question of removing the name. I found the paragraph on your research philosophy and approach quite confusing and suggest you simplify it for your readers. I don’t agree that research is the key to solving resource-related issues in low to middle income countries.

Some important details are missing in the method section including on governance. You describe reaching mutual understanding with your participants but have not addressed reflexivity and considered bias- always inherent in research but more so when you appear to be working with participants? How did you carry our member checking after each interview? Please provide clearer information on your method of analysis.

I’ve made comments in the results section. The discussion needs to be re-written completely. Unless PLoS have another structure, I suggest:

Key findings, strengths and limitations, discussion in relation to the literature, related areas for research, conclusions

I think this is an issue you feel strongly about but your roles are as researchers and you should temper what comes across as almost anger. Some of your recommendations (which shouldn’t be within the main discussion section) are greatly outwith the scope of the research.

Several errors in listing of references – please check all against PLoS requirements. I hope you will find these comments helpful and wish you well.

Reviewer #2: The manuscript highlighted the challenges experienced by nurses working in an emergence unit in Ghana. This manuscript displays knowledge of the methodical aspects of qualitative research, i.e. research design, sampling and data collection, data analysis and qualitative rigor and ethics. The participants’ voices form the themes which are further discussed in the discussion section.

My only comment is when was this study conducted? Was this before or during the pandemic that we are currently experiencing? My opinion is that if the study was conducted at the beginning/during the pandemic then the participants’ responses could be different or have more to add regarding the overcrowding and lack of resources and managerial support.

6. PLOS authors have the option to publish the peer review history of their article (what does this mean?). If published, this will include your full peer review and any attached files.

Reviewer #1: No

Reviewer #2: **Yes: **Dr Dorien Wentzel

---

## [Author Response · Author response to Decision Letter 0]

19 Jan 2021

Dear Editor

We would like to thank you sincerely for the review comments on our manuscript. We find the comments very useful and have responded to them to the best of our knowledge. We acknowledge that the comments have no doubt helped improve the quality of our manuscript. We herein provide further details by showing point-by-point feedback on how each of the comments received was addressed. For easy identification, the reviewers’ comments have been repeated while the Authors’ responses appear in red bold text.

We also seek to amend authorship. Dr. Vida Nyagre Yakong was mistakenly removed during the submission process, we, therefore, wish to add her to the authors of the said manuscript. She contributed fully to this study and meets the authorship. Thank you.

Reviewer #1: Thank you for submitting this paper which I have read with interest and enjoyment. You and your colleagues have addressed an important area with implications for practice and patient safety. I think you’ve gathered very interesting data which I’m sure will add to an important debate.

Dear Reviewer

We acknowledge and appreciate your thorough review of our manuscript. We find the comments very useful and have responded to them to the best of our knowledge. We acknowledge that the comments have no doubt helped improve the quality of our manuscript. 

Reviewer comments

I have made many comments which I hope will be helpful. Some contextualising information in the introduction section would be helpful for example how patients are triaged in the ED. 

Authors response

The facility did not have a formal triage system at the ED. Patients were generally assessed without following any triage process. The authors seek to further explore triaging among nurses in the various hospitals in the region which is our current project. 

Reviewer comments

I am concerned that the hospital may be identifiable and suggest that you anonymise it further – not just a question of removing the name.

Authors response 

Thank you. We did also observe that, we have therefore revised the study setting.

Reviewer comments

I found the paragraph on your research philosophy and approach quite confusing and suggest you simplify it for your readers. I don’t agree that research is the key to solving resource-related issues in low to middle income countries.

Authors response

Equally, we don’t disagree with you. Indeed, research is not the only form of resolving clinical issues. We, therefore, removed the sentence.

Reviewer comments

Some important details are missing in the method section including on governance. 

You describe reaching mutual understanding with your participants but have not addressed reflexivity and considered bias- always inherent in research but more so when you appear to be working with participants? 

Authors response

Research team and reflexivity has been included in the methods column 

Reviewer comments

How did you carry our member checking after each interview?

Authors response 

Member checking was done by returning the interview transcripts to participants to confirm their responses. Each participant was given the transcribed data to confirm responses by their codes. Member checking is a tool that enhances the trustworthiness or validity of a study, so the authors left this portion in the rigor column 

Reviewer comments

Please provide clearer information on your method of analysis.

Authors 

Authors have given more clear information on the analysis of data

Reviewer comments

I’ve made comments in the results section.

Authors response

Dear reviewer, we have addressed all the comments and suggestions. 

Reviewer comments 

The discussion needs to be re-written completely. Unless PLoS have another structure, I suggest:

Key findings, strengths and limitations, discussion in relation to the literature, related areas for research, conclusions

Authors response

We appreciate your suggestion on our discussion but the discussion and the manuscript structure are in line with the structure provided by PLoS One. We, therefore, wish to maintain the discussion (https://journals.plos.org/plosone/s/submission-guidelines). 

Reviewer comments

I think this is an issue you feel strongly about but your roles are as researchers and you should temper what comes across as almost anger. Some of your recommendations (which shouldn’t be within the main discussion section) are greatly outwith the scope of the research.

Authors response

Dear reviewer, we appreciate this comment greatly and have therefore removed recommendations that are not within the scope of the study. 

Reviewer comments

Several errors in listing of references – please check all against PLoS requirements. I hope you will find these comments helpful and wish you well.

Authors response 

We appreciate the thorough review of the manuscript and indeed the manuscript is greatly improved and much clearer than before. The references have been checked and changes effected.

Reviewer #2: The manuscript highlighted the challenges experienced by nurses working in an emergence unit in Ghana. This manuscript displays knowledge of the methodical aspects of qualitative research, i.e. research design, sampling and data collection, data analysis and qualitative rigor and ethics. The participants’ voices form the themes which are further discussed in the discussion section.

Authors response

Dear Reviewer, we appreciate your positive comments about our manuscript. 

My only comment is when was this study conducted? Was this before or during the pandemic that we are currently experiencing? My opinion is that if the study was conducted at the beginning/during the pandemic then the participants’ responses could be different or have more to add regarding the overcrowding and lack of resources and managerial support.

Authors response

This study was conducted earlier before COVID 19 pandemic

---

## [Editor Report · Decision Letter 1]

1 Feb 2021

“We are left with nothing to work with”; challenges of nurses working in the emergency unit at a secondary referral hospital: a descriptive qualitative study

PONE-D-20-22261R1

Dear Dr. Afaya,

We’re pleased to inform you that your manuscript has been judged scientifically suitable for publication and will be formally accepted for publication once it meets all outstanding technical requirements.

Kind regards,

Katie MacLure, PhD, MSc (dist)., BSc (1st)

Academic Editor

PLOS ONE

Additional Editor Comments (optional):

These points are minor so can be corrected during the proofing process. Please check your reference to Atakro in the Discussion section which I believe should be 23 not 22. Also where you have reference numbers covering a consecutive range present these as [23-30]. Please remove the possessive apostrophe four lines above Data Analysis "participants were not interrupted".
---

## [Editor Report · Acceptance letter]

5 Feb 2021

PONE-D-20-22261R1 

“We are left with nothing to work with”; Challenges of nurses working in the emergency unit at a secondary referral hospital: a descriptive qualitative study 

Dear Dr. Afaya:

I'm pleased to inform you that your manuscript has been deemed suitable for publication in PLOS ONE. Congratulations! Your manuscript is now with our production department. 

Kind regards, 

on behalf of

Dr. Katie MacLure 

Academic Editor

PLOS ONE